# Performance boost for bismuth telluride thermoelectric generator via barrier layer based on low Young's modulus and particle sliding

Yuxin Sun[1,8], Fengkai Guo [1,8] ✉, Yan Feng [2], Chun Li[3], Yongchun Zou[4], Jinxuan Cheng[5], Xingyan Dong[1], Hao Wu[1], Qian Zhang [5], Weishu Liu [6], Zihang Liu [3], Wei Cai[1], Zhifeng Ren [7] ✉ & Jiehe Sui [1] ✉

The lack of desirable diffusion barrier layers currently prohibits the long-term stable service of bismuth telluride thermoelectric devices in low-grade waste heat recovery. Here we propose a new design principle of barrier layers beyond the thermal expansion matching criterion. A titanium barrier layer with loose structure is optimized, in which the low Young's modulus and particle sliding synergistically alleviates interfacial stress, while the $TiTe_2$ reactant enables metallurgical bonding and ohmic contact between the barrier layer and the thermoelectric material, leading to a desirable interface characterized by high-thermostability, high-strength, and low-resistivity. Highly competitive conversion efficiency of 6.2% and power density of 0.51 W cm$^{-2}$ are achieved for a module with leg length of 2 mm at the hot-side temperature of 523 K, and no degradation is observed following operation for 360 h, a record for stable service at this temperature, paving the way for its application in low-grade waste heat recovery.

Thermoelectric (TE) devices that can convert thermal energy into electrical energy have been widely used in maintenance-free power supply systems for deep space exploration and other extreme environments[1–3]. Additionally, there is great promise for the use of TE devices in low-grade waste heat recovery since more than 60% of the energy generated through burning fossil fuels is discharged into the environment as waste heat, about half of which is low-grade[4], for which there remains a lack of effective recycling methods. The most common configuration of these TE devices is to connect p- and n-type TE legs with electrodes electrically in series and thermally in parallel[5].

The conversion efficiency ($\eta$) of TE devices is related to the dimensionless figure of merit ($ZT$) of the constituent TE materials and the connection quality[6–8]. To prevent performance degradation or device failure caused by mutual diffusion between the TE materials and the electrodes during connection and service, it is essential to introduce a diffusion barrier layer on the surface of the TE materials[9]. In the last two decades, significant breakthroughs have been made in improving the performance of TE materials[10–12]. Unfortunately, barrier layer design remains a challenge since the interface between the barrier layer and the TE material should exhibit both high strength and low resistivity

[1]National Key Laboratory for Precision Hot Processing of Metals, Harbin Institute of Technology, 150001 Harbin, China. [2]State Key Laboratory of Solidification Processing, Northwestern Polytechnical University, 710072 Xi'an, China. [3]State Key Laboratory of Advanced Welding and Joining, Harbin Institute of Technology, 150001 Harbin, China. [4]Center of Analysis Measurement and Computing, Harbin Institute of Technology, 150001 Harbin, China. [5]School of Materials Science and Engineering and Institute of Materials Genome & Big Data, Harbin Institute of Technology, 518055 Shenzhen, China. [6]Department of Materials Science and Engineering, Southern University of Science and Technology, 518055 Shenzhen, China. [7]Department of Physics and Texas Center for Superconductivity at the University of Houston (TcSUH), University of Houston, Houston, TX 77204, USA. [8]These authors contributed equally: Yuxin Sun, Fengkai Guo. ✉e-mail: fkguo@hit.edu.cn; zren@uh.edu; suijiehe@hit.edu.cn

while ensuring long-term device stability at the service temperature, and the current lack of adequate barrier layers is a bottleneck limiting the application of TE devices in waste heat recovery[13–15]. To satisfy the above-mentioned requirements, it has long been assumed that the following standards need to be met between the barrier layer and the TE material: coefficient of thermal expansion (CTE) matching to reduce interfacial stress, a certain degree of reaction at their interface to facilitate strong bonding, and work function matching to ensure low levels of contact resistivity and parasitic loss[16–19].

Bismuth telluride (BiTe) exhibits chemical stability and excellent low-temperature TE performance[20,21] and is currently the most reliable choice for low-grade waste heat recovery using TE technology. Ni has been widely used as a barrier layer for BiTe-based devices because of its similar CTE value[22,23]. However, when such a device is operated above 473 K, the Ni will continue to react with the BiTe to form brittle Ni-Te compounds, leading to an increase in both interface resistivity and CTE mismatch until cracking and failure occur, especially in the n-type joint[24–26]. Although new barrier layers that meet the CTE matching criterion, including Fe, Co-P, FeCrNi, and Ni-based alloy[27–31], have been continuously explored in recent years, they have not been demonstrated to exhibit sufficiently low contact resistivity or the ability to ensure long-term device stability above 473 K. In summary, due to the lack of ideal barrier layers, the highest temperature for long-term stable service of BiTe devices is limited to below 473 K, restricting the improvement of conversion efficiency. Therefore, it is urgent to develop a thermally stable barrier layer that exhibits both high strength and low resistivity above 473 K.

Since the stability of a TE device's interfacial mechanical structure is a prerequisite for its long-term operation, it is commonly believed that the CTE mismatch between the barrier layer and the TE material must be limited to within ±20%[32]. However, the mechanical properties of various TE materials differ greatly. Among IV–VI families with poor mechanical properties, minimal interfacial stress may lead to material fracture, so highly matched CTE values are necessary. Therefore, an additional stress buffer layer has been introduced at the interface between the barrier layer and the TE material in, e.g., PbTe-, GeTe-, and SnTe-based legs[18,33]. In contrast, TE materials with good mechanical properties can resist the large interfacial stress caused by CTE mismatch, such as in joints composed of Nb ($-7.3 \times 10^{-6}$ $K^{-1}$) and a skutterudite ($9.3–11.3 \times 10^{-6}$ $K^{-1}$)[34], Cr ($-4.9 \times 10^{-6}$ $K^{-1}$) and a half Heusler ($-11 \times 10^{-6}$ $K^{-1}$)[35,36], and Fe ($-11.8 \times 10^{-6}$ $K^{-1}$) and $Mg_3$(Bi, Sb)$_2$ ($-23 \times 10^{-6}$ $K^{-1}$)[37,38]. These results indicate that CTE matching is not the only criterion for determining the stability of a device's interfacial mechanical structure, at least during short-term testing. It must be noted that the mechanical stability of the above interfaces still needs to be tested under long-term service and thermal shock conditions. Interfacial stress is caused by the mismatch in deformation between the two sides of the interface as the temperature varies. If one side of the TE joint can produce significant recoverable deformation, the interfacial stress during cold- and hot-cycling will be effectively alleviated. This ability is directly related to a particular mechanical parameter of a material, the Young's modulus. Due to their large elastic deformation under low stress, barrier layer materials with low Young's modulus values can partly compensate for the difference in expansion or shrinkage caused by CTE mismatch, thus relieving the interfacial stress and maintaining structural stability. Therefore, a material with a low Young's modulus value still has the potential to serve as a barrier layer even if its CTE does not match that of the TE material. Among the frequently used barrier layer candidates for TE devices, Ti, Zr, and Nb possess lower Young's modulus values (116, 68, and 105 GPa, respectively), and among these, Ti has the CTE ($-8.6 \times 10^{-6}$ $K^{-1}$) closest to that of BiTe ($-14–18 \times 10^{-6}$ $K^{-1}$)[7] while also having the advantages of high specific strength, better processability, decent corrosion resistance, and low cost. Therefore, the possibility of using Ti as a barrier layer for a BiTe-based TE device is comprehensively studied here.

## Results and discussion

Briefly, two joints, Ti/p-type BiTe [Ti/(BiSbTe)] and Ti/n-type BiTe [Ti/(BiTeSe)], were fabricated using one-step sintering. In each, a reaction layer of TiTe$_2$ with a thickness in the range of a few nanometers to tens of nanometers formed between the BiTe and the Ti, realizing metallurgical bonding and ensuring high bonding strength. Due to the low sintering temperature, the Ti layer exhibits a loose structure, which results in an even lower Young's modulus value and particle sliding, therefore releasing the interfacial stress during the cooling process as shown in the schematic diagram in Fig. 1a (a 2-micron-thick layer of Ni was electroplated as a solderable layer to facilitate the connection to the electrode). The formation of the TiTe$_2$ reaction layer and the loose structure of the Ti barrier layer together ensure a decent interfacial shear strength of ~12 MPa and a tensile strength of ~10 MPa at room temperature. Additionally, the interface between TiTe$_2$ and BiSbTe (BiTeSe) has an ohmic contact with extremely low resistivity of <3 μΩ cm$^2$ and there is almost no change in contact resistivity or bonding strength after aging at 523 K for 45 days. A fabricated module incorporating BiSbTe and BiTeSe as p- and n-type legs, respectively, with Ti as the barrier layer and a leg length of 2 mm was found to exhibit considerable conversion efficiency of 6.2% and power density of 0.51 W cm$^{-2}$ at a hot-side temperature $T_h$ = 523 K and a cold-side temperature $T_c$ = 293 K, both of which are highly competitive in comparison to reported results for modules with similar leg length[20,21,39–44], as shown in Fig. 1b. More excitingly, there was no noticeable deterioration among the output properties of the fabricated module after 30 thermal shocks over a total service time of 360 h at $T_h$ of 523 K, demonstrating its service stability and reliability (Fig. 1c). Compared with the current service performance of bismuth telluride devices using Ni (or Ni-based alloys) as barrier layers, the module obtained here shows significantly better stability[27,31]. In summary, we have, for the first time, developed a highly stable BiTe-based TE device that is able to operate at 523 K for energy harvesting. Additionally, the loosely structured barrier material, which is expected to relieve the interfacial stress, can be generally applied to accelerate the design of emerging TE devices.

For the TE module studied here, laboratory sintered Bi$_{0.399}$Sb$_{1.596}$Pb$_{0.005}$Te$_3$ (BiSbTe) and commercially extruded BiTe (BiTeSe, approximate composition Bi$_2$Te$_{2.7}$Se$_{0.3}$) were selected for the p- and n-type legs, respectively, based on their relatively high TE performance. Due to the poor weldability of Ti, Ni was electroplated on the outside surface of the Ti layer to reduce the contact resistivity and increase the bonding strength, as well as to facilitate device assembly. The Ti layer surface after corroding and electroplating and a joint interface after welding are shown in Supplementary Fig. 1. The Ti/BiSbTe and Ti/BiTeSe joints exhibit extremely low contact resistivity ($\rho_c$) values of 2.5 μΩ cm$^2$ and 2.3 μΩ cm$^2$, respectively, as shown in Supplementary Fig. 2a, b, which are lower than those of other reported TE joints[25,45,46]. The room-temperature shear strength and tensile strength of the joints reach ~12 MPa and ~10 MPa, respectively, indicating that the joints can meet the service requirements (Supplementary Fig. 2c, d). Schematic diagrams of the shear strength and tensile strength measurements and optical images of the tested joints are shown in Supplementary Fig. 2e, f. Backscattered electron (BSE) images of the Ti/BiSbTe and Ti/BiTeSe joints after sintering and their corresponding compositional line profiles (Supplementary Fig. 3), together with energy dispersive spectroscopy (EDS) mapping of these joints (Supplementary Fig. 4) indicate that no obvious reactions occur at the interfaces.

Scanning transmission electron microscopy (STEM) was used to further explore the interface structure. As shown in Fig. 2a, there is an interface layer ~10 nm thick between Ti and BiSbTe. A wider Te- and Ti-rich region appears around the area with small curvature radius on the surface of BiTe (indicated by the dash-dotted outline) due to the faster reaction rate. The high-resolution TEM (HRTEM) image and the

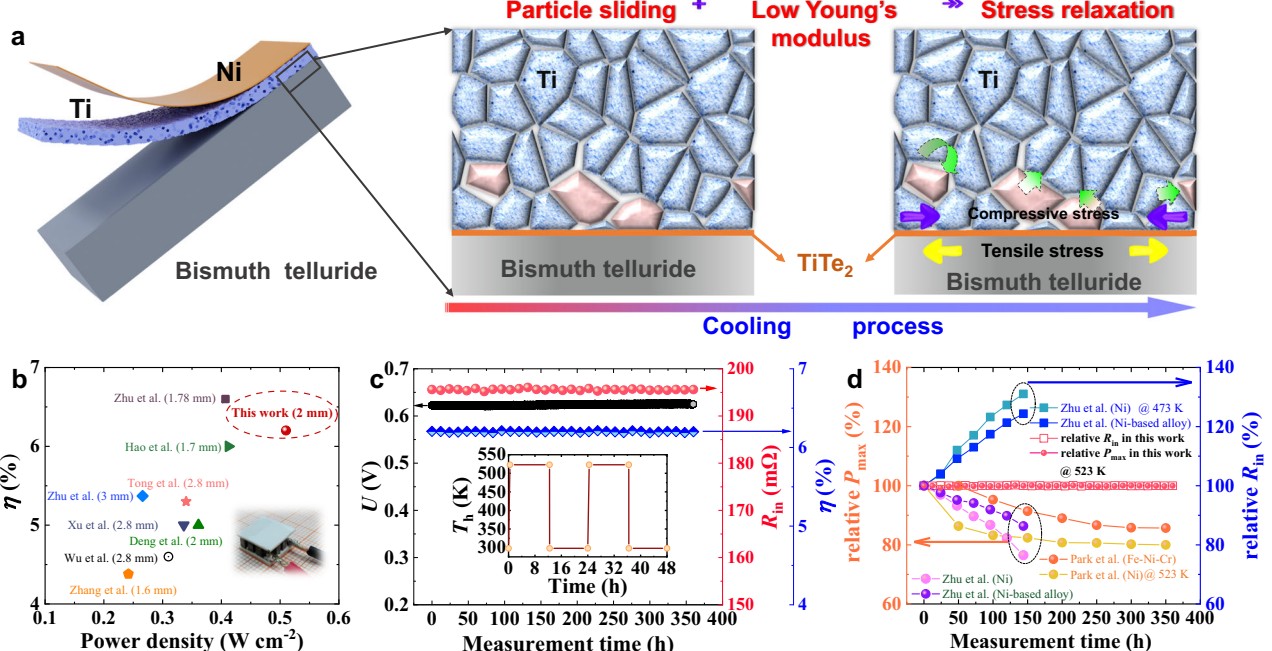

**Fig. 1 | Schematic diagram of stress release mechanism and the output performance of the module. a** Schematic diagram of interfacial stress release via a lowered Young's modulus and particle sliding. A 2-micron-thick layer of Ni was electroplated for better soldering with the electrode. **b** Comparison of conversion efficiency and power density between the BiTe module studied here and other reported modules of similar leg length[20, 21, 39–44]. Inset: photograph of the fabricated BiTe module. **c** Long-term measurement of output voltage ($U$), internal resistance ($R_{in}$), and efficiency ($\eta$) of this module at the hot-side temperature of 523 K. Inset: temperature profile of heating and cooling every 12 h. **d** Measurement time dependence of relative $P_{max}$ and relative $R_{in}$ for the module in this work compared to other bismuth telluride-based modules[27, 31].

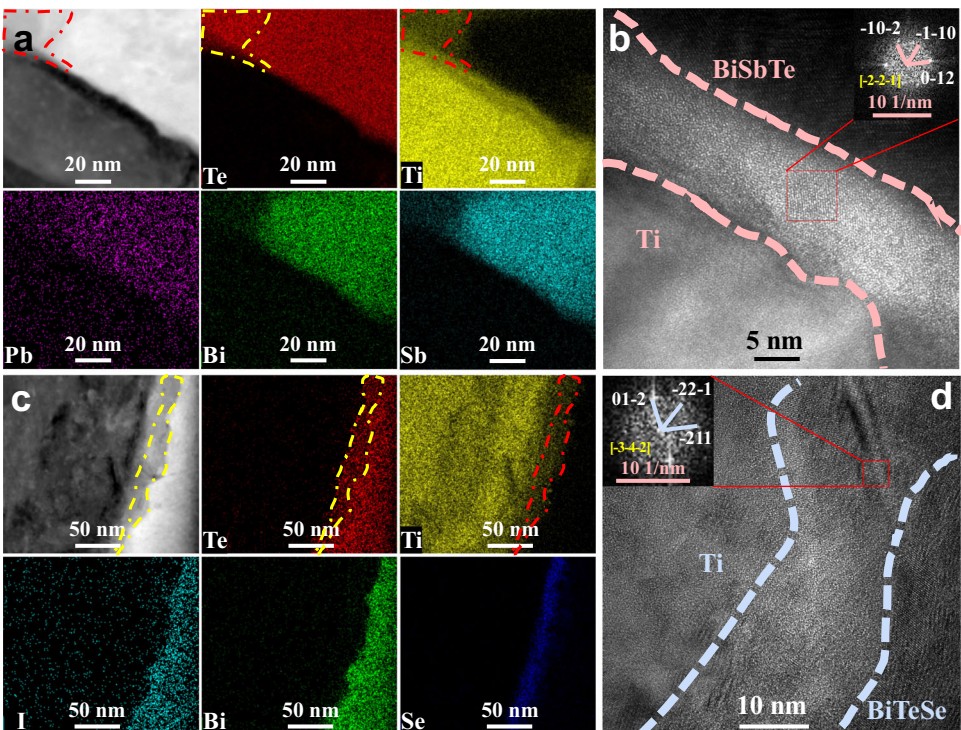

**Fig. 2 | STEM analysis of interface microstructure. a** High-angle annular dark-field (HAADF) image and EDS elemental mapping of the Ti/BiSbTe joint. **b** HRTEM image showing details of the interface between Ti and BiSbTe. Inset: corresponding FFT image. **c** HAADF image and EDS elemental mapping of the Ti/BiTeSe joint. **d** HRTEM image showing details of the interface between Ti and BiTeSe. Inset: corresponding FFT image.

corresponding fast Fourier transform (FFT) image in Fig. 2b and its inset confirm that the Te- and Ti-rich area is TiTe$_2$. In contrast, a thicker interface layer of ~30 nm consisting of two portions appears between Ti and BiTeSe (dash-dotted outline, Fig. 2c). The region near the Ti is Se-rich, while that near the BiTeSe is Te- and Ti-rich. FFT results indicate that the reactant is TiTe$_2$ in this case as well (Fig. 2d). These results imply that metallurgical bonding occurs at the joint interfaces, which ensures decent bonding strength. In addition, both interface layers consist of many disordered small grains, thus generating incoherent and disordered interfaces between TiTe$_2$ and BiTe, which is directly related to the interface contact resistivity, as detailed below.

To investigate the interfacial reaction or diffusion mechanism between Ti and BiTe, an accelerated reaction experiment, in which the Ti/p(n)-type BiTe joints were sintered at 773 K for 30 min, was carried out. As shown in Supplementary Fig. 5, interface layers ~100 μm thick appear at the interfaces of both joints. It can be seen that the diffusion direction is from the BiTe to the Ti, and the diffusion mode is grain boundary diffusion. Supplementary Fig. 5b shows an enlarged view of the selected interface area in Supplementary Fig. 5a, in which the bright regions are the Bi-rich phase. The reason for the relatively small amount of Bi-rich phase in the interface layer is that the diffusion rate of Sb and Te is greater than that of Bi[47]. The same phenomenon occurred in the n-type joint, indicating that the diffusion rate of Te and Se is also greater than that of Bi (Supplementary Fig. 5e). From Supplementary Fig. 5c, f, it can be seen that Te possesses the fastest diffusion rate among the constituent elements of both the p- and n-type BiTes for reacting with Ti, thus generating TiTe$_2$ at both interfaces. Although the diffusion rate of Se is higher than that of Bi, it is not consumed in a reaction with Ti, and it thus remains at the boundary between the Ti and the interface layer. Moreover, X-ray diffraction (XRD) patterns of cross-sections of the joints also indicate that the interface layers are TiTe$_2$, as shown in Supplementary Fig. 6. From the above, it can be seen that increasing the sintering temperature can increase the thickness of the reaction layer, which may improve the interfacial bonding strength. In addition, since the sintering temperature of 673 K is already capable of inducing interfacial reaction to take place, prolonging the sintering time may also increase the thickness of TiTe$_2$ layer. Therefore, we attempted to sinter the joints at 723 K for 5 min and at 673 K for 30 min, respectively, but the results showed a decrease in TE performance and a significant increase in contact resistivity of the n-type joint although the thickness of TiTe$_2$ layer indeed increase (Supplementary Figs. 7 and 8), which was not worth the loss. Therefore, we retained the original sintering process of 673 K and 5 min.

To study their thermal stability, p- and n-type joints sintered at 673 K were aged at 523 K in vacuum for 15, 30, or 45 days. It can be seen from Fig. 3a that the $\rho_c$ values of both joints remain quite low throughout the aging and from Supplementary Fig. 9 that there are still no observable interface layers at the interface of either joint after aging for 45 days, which may be due to the extremely slow atomic diffusion at this temperature that cannot be detected, or the atoms have not obtained enough energy to break through the diffusion barrier. The interface bonding strength (shear strength of ~12 MPa and tensile strength of ~10 MPa) of the pristine joints remains basically unchanged as well throughout aging (Fig. 3b), implying outstanding durability and stability. For comparison, Ni/BiSbTe and Ni/BiTeSe joints were also fabricated, and their $\rho_c$ values and BSE images after aging under the conditions described above are shown in Supplementary Fig. 10. In agreement with other reports[25,31,47], the n-type joint was found to exhibit a faster reaction rate, leading to a larger increase in its $\rho_c$ value with aging at 523 K. More seriously, the resulting thick interface layer significantly reduces the bonding strength of the n-type joint, causing the interface to break during the polishing process (inset, Supplementary Fig. 10f2). However, we also have to admit that due to the stronger reaction between Ni and BiTe, the shear and tensile strength

of as sintered Ni/BiTe joints are also higher than those between Ti and BiTe, as shown in Supplementary Table 1.

Due to the presence of the TiTe$_2$ interface layer in both the Ti/BiSbTe and Ti/BiTeSe joints, the contact characteristics of each are determined by the interface between the respective BiTe and the interface layer since there is almost no potential barrier between the semi-metallic TiTe$_2$ and Ti. To visually analyze the contact characteristics, TiTe$_2$ is treated as a metal here. The work function values of BiSbTe, BiTeSe, and TiTe$_2$ were measured to analyze the interface contact characteristics. As shown in Fig. 3c–e, a Schottky contact is formed at the interface for each joint, which will hinder the transmission of charge carriers and theoretically lead to a large $\rho_c$. However, the measured results show the opposite, which is possibly related to the following two reasons. First, the interface disorder shown by TEM (Fig. 2c, d) may introduce surface states in the bandgap, reducing the Schottky potential barrier[35]. Second, the heavily doped BiTe reduces the barrier width, making it easy for charge carriers to tunnel and thus unblock their transport, leading to a low contact resistivity[48]. Therefore, the $\rho_c$ values are extremely low and the I-V characteristics for the Ti/BiSbTe and Ti/BiTeSe joints both show ohmic contact (Fig. 3f).

Due to the low sintering temperature (673 K) used for assembly of these joints, the high-melting-point Ti incorporated possesses a low density of 3.4 g/cm$^3$ (77.3% of the theoretical density, denoted 673Ti). It is reasonable to consider whether this low density plays a role in the mechanical stability of the interface structure. To investigate the underlying mechanism, a comparative experiment was designed in which a prefabricated dense Ti disc (97.8% of the theoretical density, sintered at 1173 K, denoted 1173Ti) was sintered with the BiTe powders at 673 K, as shown in Supplementary Fig. 11. Obvious cracks appear within the BiTe and at the joint interface after sintering. As respectively shown in Fig. 4a, b, the temperature-dependent CTE of 673Ti is slightly lower than that of 1173Ti but its temperature-dependent Young's modulus, calculated using temperature-dependent sound velocities (Supplementary Fig. 12 and Supplementary Note 1), is significantly lower. The temperature-dependent Poisson's ratio values displayed in Supplementary Fig. 13 show the same trend of change between 673Ti and 1173Ti as their Young's modulus values. Therefore, when the joints of BiTe and Ti with different densities are cooled from 673 K to room temperature (RT), the degree of thermal contraction mismatch at the respective interfaces should be approximately the same, but because 673Ti has a lower Young's modulus, it consumes some stress through elastic deformation, while 1173Ti has poor deformability, which eventually leads to the cracking of the interface or the BiTe material. The three-dimensional (3D) finite element method was used to analyze the distribution of von Mises stresses among 673Ti (1173Ti) and BiSbTe (BiTeSe) after cooling from 673 K to RT, as shown in Fig. 4c–f. The simulation results show that, after replacing 1173Ti with 673Ti, the stress sustained by BiSbTe and BiTeSe decreases significantly from 289.5 and 251.4 MPa to 189 and 159.4 MPa, respectively, but remains an order of magnitude higher than the tensile strength of BiTe, as shown in Supplementary Fig. 14 and Supplementary Table 2. Although a gap between simulated and experimental results is to be expected, they should not differ by an order of magnitude, and such a discrepancy indicates that there may be other factors affecting the interfacial stress.

To determine other possible factors affecting the interfacial stress, the microstructural evolution of the loosely structured Ti with temperature was investigated using the Ti/BiTeSe joint as an example, as shown in Supplementary Fig. 15. It can be seen from the corroded Ti layer surface that the Ti particles are not completely densified after sintering at 673 K, which corresponds to the measured low density of the Ti layer. Surprisingly, some Ti particles near the interface with BiTeSe appear to have slid (Fig. 4g–k) after the temperature increased from RT to 573 K while no sliding can be seen among those outside the interface area (Supplementary Fig. 15c, d). Therefore, it is believed that

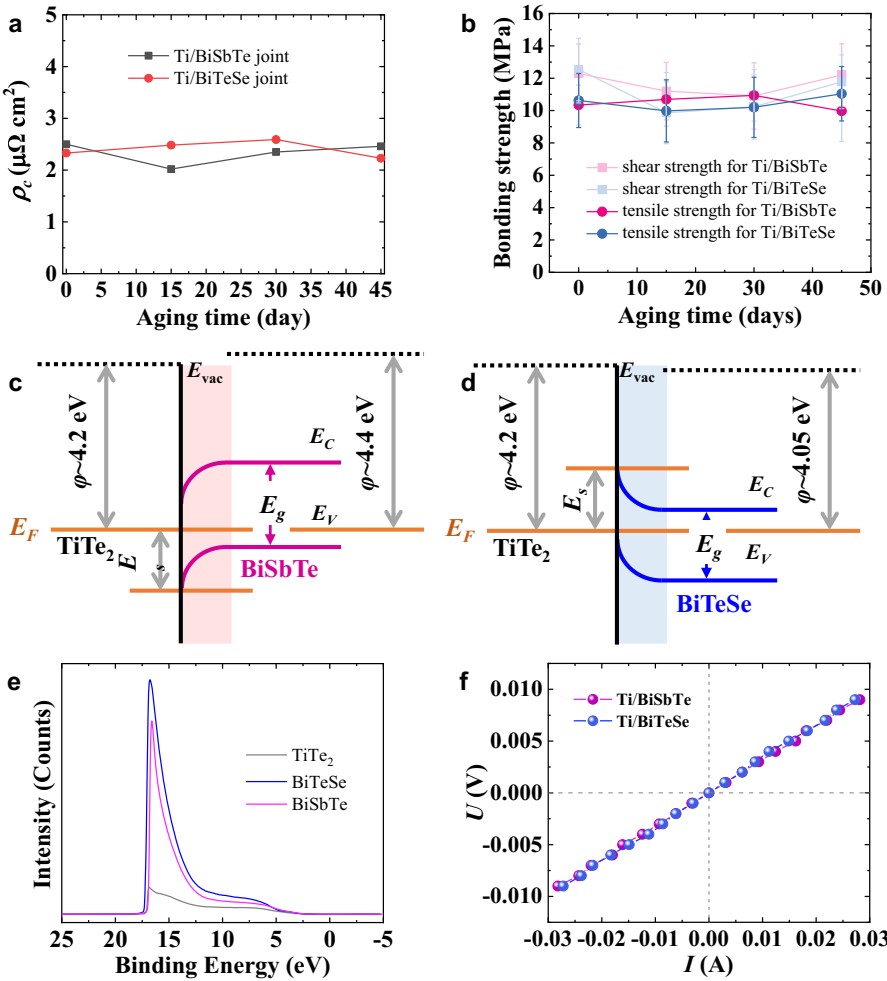

**Fig. 3 | Investigation of the interface properties of Ti/BiSbTe(BiTeSe) joints.** Changes in **a** $\rho_c$ and **b** bonding strength with aging time. Schematic illustrations of the band alignment at the interface **c** between TiTe$_2$ and BiSbTe and **d** between TiTe$_2$ and BiTeSe. **e** Ultraviolet photoelectron spectroscopy (UPS) spectra of TiTe$_2$, BiSbTe, and BiTeSe. **f** $I$–$V$ characteristics of the Ti/BiSbTe and Ti/BiTeSe joints.

the observed sliding is driven by interfacial stress caused by CTE mismatch during the heating process, a mechanism known as "splat sliding"[49–51], or more accurately called "particle sliding" in this case. In addition, the joint was further heated to 623 K (Supplementary Fig. 15e), while no significant sliding continues to occur, which may be due to that the previous sliding has already consumed most of the interfacial stresses, while the subsequent accumulated stress is no longer sufficient to drive new sliding due to the reduction of Young's modulus at high temperatures, especially for the Ti layer. After cooling to RT, it was found that the particles previously observed to have slid did not return to their original positions (Supplementary Fig. 15f), indicating the irreversibility of such sliding. Therefore, particle sliding in this case is not an elastic deformation, but rather a kind of pseudo-plastic deformation. It is worth mentioning that sliding hardly occurs within the operating temperature range from RT to 523 K (Supplementary Fig. 16a, b), and an additional sample was tested to verify this phenomenon (Supplementary Fig. 16c, d). Due to the different initial stress states at the interface, the accumulation of interfacial stress during the heating (experimental in situ heating) and cooling (natural cooling after sintering) processes is different. However, it is not feasible to directly observe the evolution of the structure near the interface from the almost stress-free state at 673 K to RT through experiments. Therefore, based on the above experimental and simulation results, we can roughly deduce the following process: during the natural cooling after sintering at 673 K, as the interfacial stress

accumulates to a certain threshold that exceeds the maximum stress that can be consumed by the elastic deformation of the materials on either side of the interface, particle sliding (pseudo-plastic deformation) will be activated in the loosely structured Ti layer to further deplete the stress. The two mechanisms thus work together to finally alleviate the interfacial stress to a level that the BiTe can withstand, ensuring the mechanical stability of the interface structure.

To verify the applicability of BiTe devices prepared using the studied barrier layer for service at temperatures above 473 K, a 7-pair module with $10 \times 10$ mm$^2$ ceramic substrates on both top and bottom was fabricated. Finite element simulation was applied to determine the influence of the ratio of the cross-sectional area of the n-type leg to that of the p-type leg ($r_{NP}$) and that of the ratio of the TE leg height ($H$) to the total cross-sectional area of a TE leg pair ($H/A_{pn}$, $A_{pn} = A_p + A_n$) on output power and conversion efficiency, as shown in Fig. 5a, b, respectively. The optimized $r_{NP}$ value is clearly 1, which greatly facilitates module assembly. To balance the output power and conversion efficiency, the cross-sectional size of the TE leg was set at $1.6 \times 1.6$ mm$^2$ and the height was set at 2 mm (the most commonly used height in industry), i.e., $H/A_{pn}$ is equal to 0.39. Properties of the constituent TE materials are shown in Supplementary Fig. 17. It should be emphasized that, based on our calculations considering the temperature-dependent resistivity and thermal conductivity of 673Ti (Supplementary Fig. 18), the slightly lower electrical conductivity and thermal conductivity resulting from the use of the loosely structured Ti layer

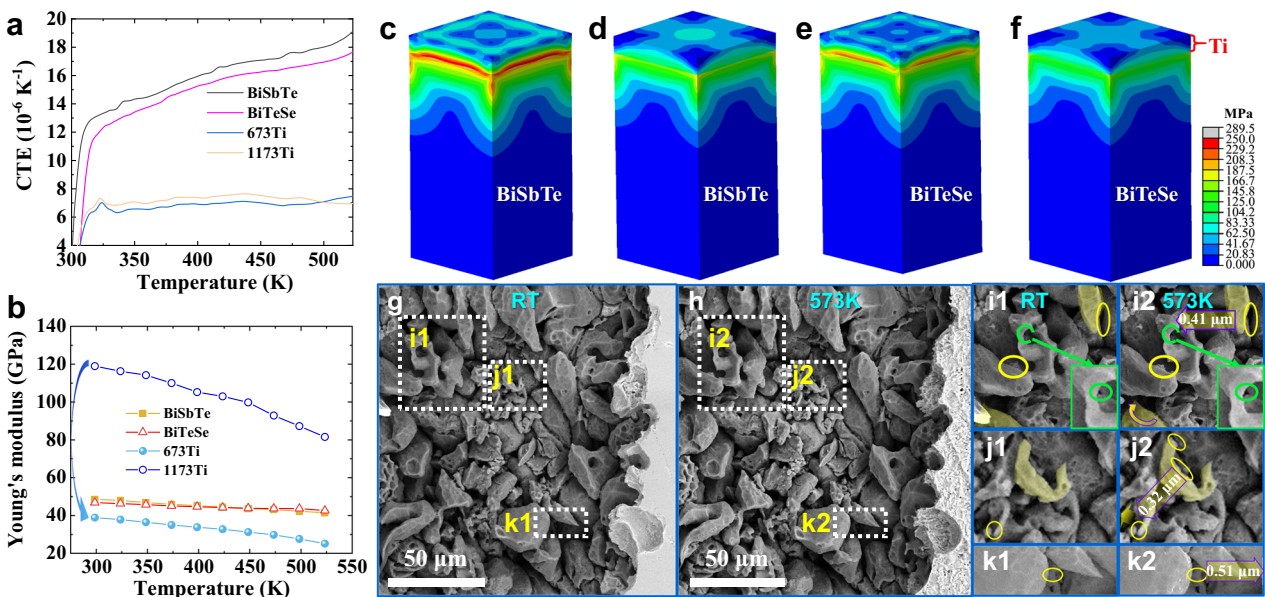

**Fig. 4 | Interfacial stress analysis and structure evolution of loose Ti layer. a** CTE and **b** Young's modulus of BiSbTe, BiTeSe, 673Ti, and 1173Ti. Distribution of von Mises stresses in **c** 1173Ti/BiSbTe, **d** 673Ti/BiSbTe, **e** 1173Ti/BiTeSe, and **f** 673Ti/BiTeSe joints after cooling from 673 K to 323 K, as obtained by finite element simulation (volume of BiSbTe or BiTeSe is 1.6 × 1.6 × 3 mm³, and Ti is 250 µm thick). SEM images of the corroded Ti layer surface at **g** RT and **h** 573 K. **i1–k1**, **i2–k2** are magnified views of the corresponding particle sliding areas in **g** and **h**, respectively, for comparison.

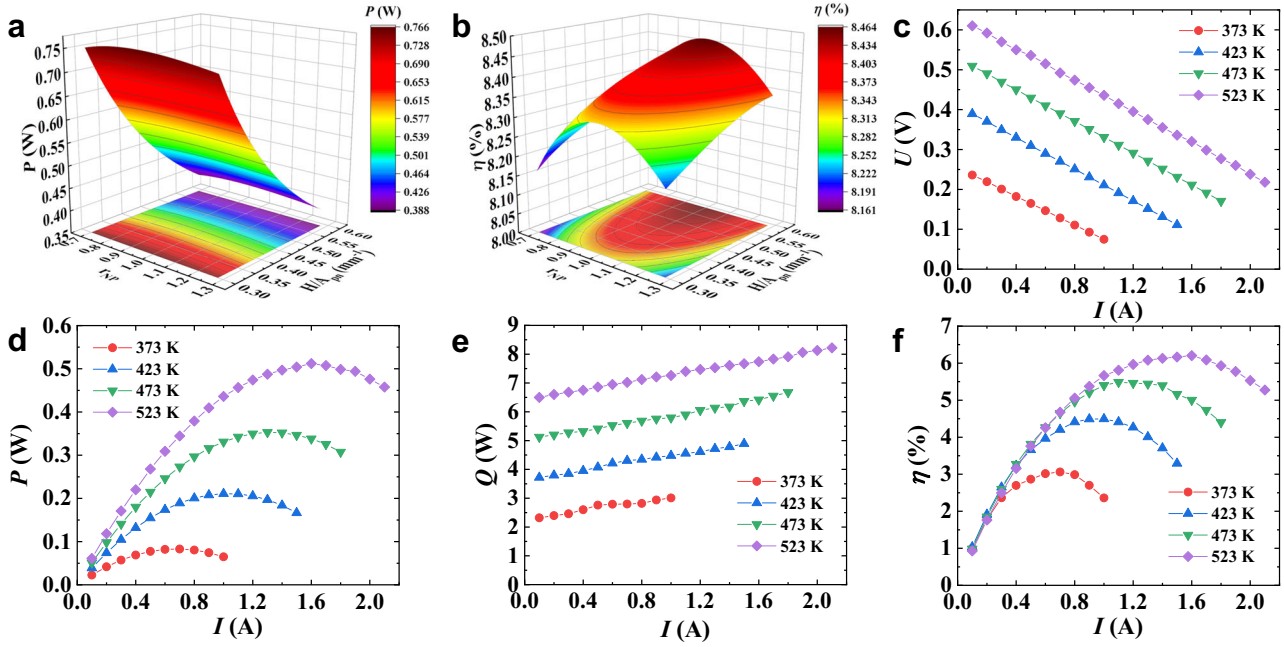

**Fig. 5 | Performance of bismuth telluride TE module.** Optimization of geometric dimensions for **a** output power and **b** conversion efficiency at $T_h$ = 523 K and $T_c$ = 293 K. **c** Output voltage, **d** output power, **e** heat fluence, and **f** conversion efficiency of the module as a function of current at different hot-side temperatures with a fixed cold-side temperature of 293 K.

only negligibly reduces the conversion efficiency of the module by 0.015%. Homemade equipment was used to characterize the performance of the module as a function of current at different hot-side temperatures, and results are shown in Fig. 5c–f. The $U$-$I$ curves shown in Fig. 5c exhibit a good linear relationship. When $T_h$ is 523 K and $T_c$ is 293 K, a maximum output power ($P$) of 0.51 W (Fig. 5d) and a maximum $\eta$ of 6.2% (Fig. 5f), both of which are highly competitive, are achieved at a current of ~1.6 A. The heat flow ($Q$) in Fig. 5e shows an upward trend with current. The measured output performance is inferior to the

calculated values (Fig. 5a, b) because the actual temperature difference under which the module operated was slightly smaller than that detected by the thermocouples, and at the same time, the measured heat flow at the cold end is higher than that calculated due to the radiative heat leakage from the heater. To verify the durability of the module, a long-term assessment with 30 thermal shocks at $T_h$ of 523 K over 360 h was implemented. As shown in Fig. 1c, there was no attenuation of module performance during the assessment, demonstrating excellent stability. This is the first time that a BiTe device has

been demonstrated to reliably serve at 523 K for such a long time without performance degradation, setting a new record and strongly promoting the application of such devices in the field of power generation.

In conclusion, we systematically studied the interface characteristics of Ti/BiTe joints. The contact resistivity of Ti/BiSbTe(BiTeSe) is less than $3\,\mu\Omega\,cm^2$ and the bonding strength remains above 10 MPa even after aging at 523 K for 45 days. The reaction at the joint interface generates a layer of $TiTe_2$ with a thickness ranging from a few nanometers to tens of nanometers, realizing metallurgical bonding. Although the interface between $TiTe_2$ and p- or n-type BiTe should show a Schottky contact from the perspective of work function alone, the tunneling effect and interface disorder endow an ohmic contact. The combined effect of a lowered Young's modulus and particle sliding in the loosely structured Ti barrier layer reduces the interfacial stress during the cooling process after sintering, thus ensuring the mechanical stability of interface structure. A fabricated module incorporating this Ti barrier layer exhibits both high efficiency and high power density and shows excellent stability. This study promotes the application of BiTe in low-grade heat recovery and provides a new perspective for the design of barrier layers for use with other TE materials.

## Methods

### TE module fabrication

Bi (99.999%), Sb (99.999%), Te (99.999%), and Pb (99.999%) were weighed for the synthesis of p-type $Bi_{0.399}Sb_{1.596}Pb_{0.005}Te_3$. These raw materials were sealed in evacuated quartz ampoules, melted at 1073 K for 10 h, and then cooled to room temperature in a furnace. The ingots were high-energy ball-milled into powders using a SPEX-8000M ball mill. The powders were loaded into a graphite die and sintered into bulk using a spark plasma sintering system at 673 K under 80 MPa for 5 min. Commercially extruded BiTe (approximate composition $Bi_2Te_{2.7}Se_{0.3}$) was selected for the n-type TE legs. Ti powder (99.5%) or Ni powder (99.98%) was used to fabricate the barrier layer for each TE material. The joint was prepared by one-step sintering in the order of Ti/BiTe/Ti or Ni/BiTe/Ni for comparison. All joints were prepared using the same process, in which a BiTe alloy with Ti or Ni barrier layers was placed in a vacuum quartz tube and then annealed at 523 K in a muffle furnace for 15, 30, or 45 days. The nickel plating solution was purchased from Beichen Hardware Technology Co., Ltd. Firstly, a mixed acid of hydrochloric acid and nitric acid (concentrated hydrochloric acid 100 mL/L, concentrated nitric acid 250 mL/L) was used to corrode the surface of bismuth telluride to increase surface roughness and remove oxide film. The corrosion time was 3 min. After ultrasonic cleaning, hydrochloric acid (concentrated hydrochloric acid 80 mL/L) was used for activation. The electroplating was carried out in a constant temperature water bath at 55 °C for 5 min. The current density is $1\,A/dm^2$. Each TE leg was welded onto copper-clad ceramic substrates using PbSnAg solder at 573 K for the hot side and SnBi solder at 423 K for the cold side.

### Characterization methods

The crystal structure of the samples was examined using X-ray diffraction (Empyrean, Holland). Scanning electron microscopy (SEM) with an energy-dispersive spectrometer (Merlin Compact, Germany) was used to analyze their morphology and microstructure. High-temperature SEM analysis was performed on a MIRA3 (TESCAN). Transmission electron microscopy (TEM) (Talos F200X, USA) was employed to characterize the interfacial microstructure of the samples. Electrical conductivity and Seebeck coefficient were measured simultaneously using a ZEM-3 machine. The thermal conductivity was calculated based on the measured thermal diffusivity (LFA 457, Netzsch), density, and specific heat. The contact resistivity was measured using the four-probe method and the current was set as 0.1 A. A

mechanical property test platform was employed to test the shear strength, compressive strength, and tensile strength and the load speed was 0.005 mm/min. The error bar of strength in is obtained by measuring five sets of samples. The coefficient of thermal expansion was measured using a thermal mechanical analyzer (TMA 402, Netzsch). The work function was measured using ultraviolet photoelectron spectroscopy (Thermo Fisher ESCALAB 250Xi). Temperature-dependent transverse and longitudinal sound velocities were measured using an ultrasonic testing instrument (OmniScan SX, Olympus). The commercial finite element analysis software COMSOL Multiphysics was used to optimize the geometric size of the TE module. A finite element analysis solver (ABAQUS) was used for stress simulation. The output power and conversion efficiency of the TE module were measured using homemade test equipment.

### TE module aging test

Throughout the entire testing process, the cold side of the module was maintained at 293 K. The hot side was first heated to 523 K within 25 min, maintained for 12 h, and then cooled to room temperature within 1 min, and this cycle was repeated 30 times. The output voltage was continuously evaluated every 12 h and the conversion efficiency and internal resistance were measured at the beginning of each heating period.

## Data availability

All data generated or analyzed during this study are included in the published article and its Supplementary Information. The data that support the findings of this study are available from the corresponding author upon reasonable request.

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

## Acknowledgements

This work was supported by the National Natural Science Foundation of China (Nos. 52130106 (J.S.), 52101247 (F.G.), and 52271206 (W.C.)) and the Heilongjiang Touyan Team Program (J.S.). We thank Troy J. Christensen of the Texas Center for Superconductivity for his discussion and advice.

## Author contributions

Y.S., F.G., and J.S. initiated the concept and established the experimental scheme. Y.S., Y.F., C.L., and Y.Z. synthesized the samples and performed the thermoelectric characterizations. Y.S., X.D., and H.W. established the model for finite element analysis. Y.S., J.C., and Q.Z. measured the module performance. Y.S., F.G., W.L., Z.L., W.C., Z.R., and J.S. analyzed the results. Y.S., F.G., Z.R., and J.S. completed the manuscript.

## Competing interests

The authors declare no competing interests.
