## [Peer Review File · Nature Communications]

Performance Boost for Bismuth Telluride Thermoelectric Generator via Barrier Layer Based on Low Young's Modulus and Particle SlidingREVIEWER COMMENTS

Reviewer #1 (Remarks to the Author):

In this manuscript, Yuxin Sun et al., have successfully fabricated a thermoelectric generator with Ti/p-type BiTe [Ti/(BiSbTe)] and Ti/n-type BiTe [Ti/(BiTeSe)] with an interface layer of Ti proving advantageous for improving device efficiency as well as their mechanical property. The formation of the TiTe₂ reaction layer while fabricating the legs leads to improvement in the interfacial strength and the tensile strength and has not deteriorated the contact resistance. The combined mechanical and electrical properties improved the device's stability for 30 thermal shocks at Th of 523 K over 360 hours. Overall, they have achieved an efficiency of 6.2%, which is moderate as both the stability and efficiency have been improved. Overall, from a research article perspective, the authors have written the manuscript nicely, and the result interpretation is good in the discussion part. I recommend this manuscript for publication in Nature Communications with minor revision. However, I also have some queries that need to be addressed before considering it for publication.

1. Why not use "Co" which is cheaper than "Ti" with a Young modulus of ~211 GPa and highly conductive than "Ti" ? any comments on this.
2. In both p and n-type material BiSbTe and BiTeSe, the bonding strength between the elements is different, as discussed in lines 185-198, so in the case of Ti as a diffusion layer, can we control the diffusion layer thickness as in the case in both the legs it can vary due to their bonding properties. The diffusion layer thickness can also be optimized for both legs to achieve a high efficiency.
3. In the discussion part (Lines 220-230), the formation of the Schottky potential barrier was discussed, but the comparison of p and n-type has a huge difference in work function(Φ) values for BiSbTe (4.40 eV) and BiTeSe (4.05 eV) of almost 0.35 eV; no discussion regarding this. Can it influence the transport property?.
4. In lines 242-244, the density of the incorporated Ti is given, but the optimized density values are not given, as in case one can see a lower density than what was mentioned can still provide better thermal and electrical stability. So, the possible density values can be interpreted to give a conclusive value for obtaining high performance.
5. In Figure S15C, the thermal conductivity increases beyond ~375 K for n-type, which has led to a decreasing trend of zT after ~375 K; why was no effort made to optimize zT equivalent to the p-type material to achieve better device efficiency?
6. From a comparison perspective, the device stability can be compared with other devices reported in that temperature range to give a clear idea of where the work stands as exceptional.

Reviewer #2 (Remarks to the Author):

In this work, a BiTe-based TE module with the conversion efficiency of 6.2% and power density of 0.51 W cm⁻² is achieved by using Ti as the barrier layer, showing good service stability and reliability. However, there are still many questions that have not been well explained. Thus, I encourage the authors to make major revisions. Please find the comments below:

1. After long-time aging, why did not the Ti layer and BiTe material continue to react and the TiTe₂ layer become thicker?

2. Why does the TiTe₂ layer not affect the strength of TE leg? The cracks will not occur in the TiTe₂ layer?
3. Does the loose Ti layer possess enough strength? How to ensure the leg does not break inside the Ti layer during the fabrication and service?
4. The authors claimed that the particle sliding occurred in the Ti layer within the temperature range from RT to 573 K, but the sliding seems imperceptible in the SEM images (Fig.4).
5. Why did not the particles continue to slid from 573 K to 623 K?
6. Does the particle sliding occur in other metal layers? such as Zr or Nb, which possess lower Young's modulus than Ti. Why did not the authors choose the two metals as the barrier layer?
7. In the manuscript, the measured output performance is inferior to the calculated values because the smaller actual temperature difference. Maybe the authors can increase the hot side temperature and measure the output performance under the same temperature difference with the calculation.
8. There is a textual error in the Fig. S7. The (b) and (d) should be the Ti/BiTeSe instead of Ti/BiSbTe.

Reviewer #3 (Remarks to the Author):

This work aims to adopt low Young's modulus value materials as barrier layer materials as they will result in large elastic deformation under low stress and thus relieving the interfacial stress and maintaining structural stability, even if its CTE does not match that of the TE materials. However, the interfacial stress that leads to the material fracture or the device failure is three dimensional which is more complicated than the one-dimensional Young's modulus value in this work although it is helpful to choose barrier materials. There's still some questions need to be solved before the manuscript can be accepted.

1. For commercial Bi₂Te₃ devices modules, Ni/Ni-based materials are applied as barrier materials, however, the Young's modulus value of Ni is ~206 GPa which is much larger than the Ti used in this work but close CTE to BiTe. Besides, it is the brittle Ni-Te compound that leads to the interface resistivity and CTE mismatch and thus final crack and failure occur above 473 K. So it is the react compound that determines the stability of the interface rather than the Young's modulus values of the barrier materials.
2. Please clearly explain how the particle sliding decrease the interfacial stress in the manuscript as Fig. 1a has not shown the mechanical process.
3. Fig. S2 displayed the contact resistivity of as-sintered Ti/BiSbTe and Ti/BiTeSe joints and also the shear strength and tensile strength of the two joints at room temperature. How about the result for Ni/BiSbTe(Se)? Or is there any reported data for compare?
4. The authors claimed that it is the low Young's modulus value that leads to large elastic deformation under low stress, but in the discussion section few evidence is provided, please supply more proofs at this point.

October 27, 2023

Reviewer #1 (Remarks to the Author):

In this manuscript, Yuxin Sun et al., have successfully fabricated a thermoelectric generator with Ti/p-type BiTe [Ti/(BiSbTe)] and Ti/n-type BiTe [Ti/(BiTeSe)] with an interface layer of Ti proving advantageous for improving device efficiency as well as their mechanical property. The formation of the TiTe₂ reaction layer while fabricating the legs leads to improvement in the interfacial strength and the tensile strength and has not deteriorated the contact resistance. The combined mechanical and electrical properties improved the device's stability for 30 thermal shocks at Th of 523 K over 360 hours. Overall, they have achieved an efficiency of 6.2%, which is moderate as both the stability and efficiency have been improved. Overall, from a research article perspective, the authors have written the manuscript nicely, and the result interpretation is good in the discussion part. I recommend this manuscript for publication in Nature Communications with minor revision.

However, I also have some queries that need to be addressed before considering it for publication.

1. Why not use “Co” which is cheaper than “Ti” with a Young modulus of ~211 GPa and highly conductive than “Ti” ? any comments on this.

Response: Thanks for your kind reminder. Co, with its CTE and Young's modulus very close to Ni, is indeed a potential choice for a barrier layer. It is also true that there are some previous reports on Co as well as Co-P alloy as barrier layers. For the Co-P/BiTe joints, the contact resistivity is as high as ~28 $\mu\Omega\text{ cm}^2$ and ~55 $\mu\Omega\text{ cm}^2$ for p-type and n-type interfaces respectively, and they increase to ~72 $\mu\Omega\text{ cm}^2$ and ~79 $\mu\Omega\text{ cm}^2$ after annealing at 423 K for 15 days (DOI: 10.1021/acsami.9b22853; DOI:10.1007/s11664-018-6633-7). For the Co/p-BiTe joint, there is no increase in the thickness of the reaction layer after annealing at 473 K for 32 hours (DOI: 10.1021/acsami.2c10227). However, there is also report of severe interfacial reactions between Co and p-type BiTe after aging at 573 K for 2 days (DOI: 10.15541/jim20200126).

Given that the interface between n-type BiTe and the barrier layer is generally not as stable as that of the p-type one, we have previously sintered the Co/BiTeSe joint using the same sintering process of Ti/BiTe joint to test its applicability as a barrier layer. As shown in Fig. R1, a high contact resistivity of up to 23.1 $\mu\Omega\text{ cm}^2$ was obtained. What's more, the BSE image illustrates an obvious reaction layer with a thickness of 1.5 μm (Fig. R1b). Such a large contact resistivity indicates that it is not a good barrier, even if the interface stability has not yet been assessed.

Fig. R1. (a) Contact resistivity curve and (b) BSE image of Co/BiTeSe joints sintered at 673 K for 5 minutes.

2. In both p and n-type material BiSbTe and BiTeSe, the bonding strength between the elements is different, as discussed in lines 185-198, so in the case of Ti as a diffusion layer, can we control the diffusion layer thickness as in the case in both the legs it can vary due to their bonding properties. The diffusion layer thickness can also be optimized for both legs to achieve a high efficiency.

Response: Thanks for your valuable suggestion. The thickness of the TiTe_2 layer can be tuned by changing the sintering temperature. We have attempted to increase the thickness of the reaction layer by increasing the sintering temperature to increase the interfacial bonding strength. However, this resulted in a deterioration in the TE performance of both p-type and n-type BiTe and an increase in ρ_c of Ti/BiTeSe joint, as shown in Fig. R2 and Fig. R3. Specially, for n-type interface, severe interfacial reaction (the thickness of reaction layer grows to several hundred nanometers) increases the contact resistivity to $40.6 \mu\Omega \text{ cm}^{-2}$, which will definitely deteriorate the device output performance. When the sintering temperature is lowered, even if the ρ_c values do not change, the TE performance also has significantly decreased (Fig. R2). Moreover, the thickness of the interfacial reaction layer is less than 30 nanometers when the joint is sintered at 673 K. It is conceivable that at lower sintering temperatures, the interfacial reaction will be weaker and the interfacial bonding strength will be lower. At the same time, the shear strength of Ti layer sintered at 673 K is only 11.2 MPa, it will be further reduced if sintered at lower temperature due to the lowered density.

Based on the above analysis, when the sintering temperature is 673 K, the performance of TE materials is the best. At the same time, the interface contact resistivity is relatively low, and the bonding strength is also adequate for module fabrication and service. Therefore, this is a relatively suitable sintering temperature. We have supplemented these results, as well as the analysis, in the manuscript (lines 200-206) and supporting information. Thanks again for your suggestion.

Fig. R2. (a1) - (d1) The thermoelectric properties of $\text{Bi}_{0.399}\text{Sb}_{1.596}\text{Pb}_{0.005}\text{Te}_3$ sintered at 623 K, 673 K, and 723 K for 5 minutes. (a2) - (d2) The thermoelectric properties of extruded n-type BiTe re-sintered at 673 K and 723 K for 5 minutes.

Fig. R3. Contact resistivity curves (a) (c) and BSE images (b) (d) of Ti/BiSbTe and Ti/BiTeSe joints sintered at 723 K for 5 minutes.

3. In the discussion part (Lines 220-230), the formation of the Schottky potential barrier was discussed, but the comparison of p and n-type has a huge difference in work function(Φ) values for BiSbTe (4.40 eV) and BiTeSe (4.05 eV) of almost 0.35 eV; no discussion regarding this. Can it influence the transport property?

Response: Thanks for your comment. The work function refers to the minimum energy that must be provided to cause an electron to escape from a solid surface to vacuum. For metals and semiconductors, the work function (ϕ) is equal to the difference between the energy of a stationary electron in vacuum (E_0) and the Fermi level (E_F) of the material. It can be said that the magnitude of the work function is directly related to the

Fermi level.

For semiconductors of similar composition, differences in the type of conduction and carrier concentration can lead to huge differences in work function, e.g., the difference of ϕ between n-type GaAs and p-type GaAs reaches 1.15 eV with similar carrier concentration of $\sim 10^{16} \text{ cm}^{-3}$. Therefore, it is normal that the work functions of BiSbTe (4.40 eV) and BiTeSe (4.05 eV) to be different because they have different compositions, different conduction types and different carrier concentrations. The work function is a representation of the Fermi level of a material, rather than a parameter that affects the transport properties.

As for the interfacial charge transport properties, the work function is only one of the influencing factors. The carrier concentration of the semiconductor itself, the arrangement of atoms at the interface, and the surface state all affect the transport properties. We have explained this in the main text, and the measured results also show that the interface behaves as an ohmic contact rather than a Schottky contact as inferred only from the work function.

4. In lines 242-244, the density of the incorporated Ti is given, but the optimized density values are not given, as in case one can see a lower density than what was mentioned can still provide better thermal and electrical stability. So, the possible density values can be interpreted to give a conclusive value for obtaining high performance.

Response: We strongly agree with your idea and have considered it before. From the perspective of TE performance, as shown in Fig. R2, 673 K is the optimal temperature for obtaining high-performance BST. When the sintering temperature is above or below 673 K, the thermoelectric transport properties of the Pb-doped BST changed due to factors such as changes in grain size and volatilization of elements, thus leading to a lower ZT .

From the perspective of Ti, the Ti layer sintered at 673 K has a shear strength of only 11.2 MPa (Table S2). Reducing the density of Ti by lowering the sintering temperature will lead to a further degradation of the mechanical properties of the Ti layer, thus deteriorate the machinability of the joint. We also tried to reduce the particle size of the Ti powder to increase the density and improve the conductivity and mechanical strength. However, due to the high activity of Ti, the smaller the particle size of Ti powder, the lower the purity. The presence of oxide films or other impurities on the surface can make it difficult for metallurgical bonding to occur between the powders at 673K, ultimately preventing the formation of a bulk material.

Overall, the density of the current Ti layer is relatively suitable. Changing the sintering process to reduce or increase its density will more or less deteriorate the thermoelectric or mechanical properties of the thermoelectric leg.

5. In Figure S15C, the thermal conductivity increases beyond $\sim 375 \text{ K}$ for n-type, which has led to a decreasing trend of zT after $\sim 375 \text{ K}$; why was no effort made to optimize zT equivalent to the p-type material to achieve better device efficiency?

Response: Thanks for your kind reminder. Before we design the barrier, the first thing that comes to mind is the selection of high-performance thermoelectric materials, especially n-type materials, as their performance is far inferior to p-type materials. However, the performance improvement of n-type BiTe is currently a big challenge and a goal that the entire thermoelectric community is constantly pursuing.

There are four main preparation methods for n-type BiTe, including zone melting, extrusion, ball milling combined with sintering, and hot deforming.

Firstly, zone melted n-type BiTe has decent TE performance and can be produced in large quantities easily. However, its mechanical property is very poor and is highly susceptible to fracture. Therefore, it is unsuitable for power generation applications that need to cope with thermal stresses at large temperature differences.

Secondly, extruded n-type BiTe not only has good TE performance, but also shows high mechanical performance, making it a suitable candidate, which is the reason why it was chosen as the n-type material in this work. However, its further performance optimization is a great challenge and one that industry and various laboratories are trying to solve. The extrusion method is very demanding on equipment and the cost of trial and error is very high, and our laboratory does not have the relevant experimental conditions for the time being.

Thirdly, in terms of polycrystalline n-type BiTe prepared by high energy ball milling, carrier concentration modulation, band structure modulation, construction of nanostructures, and many other methods have been used to improve their performance. However, the mainstream average ZT from 300 to 523 K prepared by this method is currently about the same as that of extruded samples, e.g., iodine doped samples reported in Hao's work (DOI: 10.1021/acsami.8b06533), as shown in Fig. R4a. In addition, the power factor obtained by the ball milling method is much lower than the extrusion method due to smaller grains and weaker orientation, which is not conducive to the output power.

Another effective way to improve the performance of polycrystalline samples is hot deformation (HD), but its high-performance direction is perpendicular to the pressure direction. To sinter the Ti layer, it is necessary to cut the sample in this direction first, which will be complex and cause huge waste, as shown in Fig. R4b.

Our group is skilled in using ball milling and sintering methods. We have also done a lot of work to improve the performance of n-type BiTe, but the performance obtained so far is close to that of extruded samples, and the power factor is lower. Therefore, we chose extruded samples in this work. Continuing to improve the performance of n-type BiTe, regardless of the method used, is the direction we will continue to strive for.

Fig. R4. (a) The ZT of $\text{Bi}_2\text{Te}_{2.6925}\text{Se}_{0.3}\text{I}_{0.0075}$ and extruded BTS sample. (b) The schematic diagram of reasons why Ti is not suitable as a barrier layer for samples after HD.

6. From a comparison perspective, the device stability can be compared with other devices reported in that temperature range to give a clear idea of where the work stands as exceptional.

Response: Thanks very much for your suggestions, which will significantly enhance the advantages of this work! We have added the comparison about device stability in **Fig. 1d**. The corresponding description has also been added into lines 137-139 of the revised manuscript as follows.

“Compared with the current service performance of bismuth telluride devices using Ni (or Ni-based alloys) as barrier layers, the module obtained here shows significantly better stability^{27, 31}.”

Fig.1 a. Schematic diagram of interfacial stress release *via* a lowered Young's modulus and particle sliding. A 2-micron-thick layer of Ni was electroplated for better soldering with the electrode. **b.** Comparison of conversion efficiency and power density between the BiTe module studied here and other reported modules of similar leg length^{20-21, 39-44}. Inset: photograph of the fabricated BiTe module. **c.** Long-term measurement of output voltage (U), internal resistance (R_{in}), and efficiency (η) of this module at the hot-side temperature of 523 K. Inset: temperature profile of heating and cooling every 12 h. **d.** Measurement time dependence of relative P_{max} and relative R_{in} for the module

in this work compared to other bismuth telluride-based modules^{27, 31}.

In summary, thanks again for your suggestions, which have significantly improved the quality of this manuscript and we have also benefited from the discussion with you.

Reviewer #2 (Remarks to the Author):

In this work, a BiTe-based TE module with the conversion efficiency of 6.2% and power density of 0.51 W cm⁻² is achieved by using Ti as the barrier layer, showing good service stability and reliability. However, there are still many questions that have not been well explained. Thus, I encourage the authors to make major revisions. Please find the comments below:

1. After long-time aging, why did not the Ti layer and BiTe material continue to react and the TiTe₂ layer become thicker?

Response: Thanks for your careful reading. In solids, diffusion is often the only way of matter transfer. The formation and thickening of TiTe₂ layer are both based on atomic diffusion. According to the Arrhenius equation, $\ln D = \ln D_0 - Q/RT$, where D is diffusion coefficient, D_0 is the diffusion constant, Q is the activation energy, R is gas constant, and T is the absolute temperature. We can see that $\ln D$ has a linear relationship with $1/T$, i.e., the higher the temperature, the greater the diffusion coefficient of the substance. Therefore, when the temperature is low enough, diffusion is extremely slow and difficult to detect even throughout the service life of TE device. In line with our experimental results, the thickness of TiTe₂ layer is about tens of nanometers after sintering at 673 K for 5 min, while it increased to a few microns after sintering at 773 K for 30 min (Fig. S5). In addition, diffusion requires atoms to escape from their equilibrium positions, which requires an external source of sufficiently high energy, i.e., the activation energy. After aging at 523 K for 45 days (Fig. S9), no changes in the interface could be observed in the SEM, and the contact resistivity and bond strength did not change significantly, indicating that either the rate of atomic diffusion at this temperature is so low that we cannot detect it, or that the atoms can't break through their diffusion energy barriers and diffusion can't take place.

Thoroughly understanding the interface diffusion and reaction situation at this temperature is a systematic work, which is also part of our ongoing work, which of course encompasses more than just the interface between Ti and BiTe. We have added the possible reasons why the thickness of TiTe₂ did not continue to increase in lines 217-219 in the main text, "which may be due to the extremely slow atomic diffusion at this temperature that cannot be detected, or the atoms have not obtained enough energy to break through the diffusion barrier".

2. Why does the TiTe₂ layer not affect the strength of TE leg? The cracks will not occur in the TiTe₂ layer?

Response: Thanks for your comment.

For the first question, why does the TiTe₂ layer not affect the strength of TE leg? Here, we discuss tensile and shear strength separately.

Tensile strength

The tensile strengths of BiSbTe and BiTeSe are all about 10 MPa (Table S2). For the joints, the test results were also about 10 MPa, as shown in Fig. S2. However, since the

fracture occurring inside the TE material, the tensile strength of the joint should be higher than the test value.

Shear strength

For the TE materials themselves, the shear strength is more than 20 MPa (Table S2). However, the interfacial shear strength is about 12 MPa (Fig. S2) for both the p-type and n-type joints.

The reason for the lower shear strength at the interface is that there is a large residual stress at the interface due to the mismatch in thermal expansion between the two sides of the interface. The residual stress mainly acts in the in-plane direction, thus having a significant impact on shear strength.

In summary, for a TE leg, the part with the lowest tensile strength is the TE material itself, while the part with the lowest shear strength is the Ti/BiTe interface. In other words, **the interface reaction layer TiTe_2 has no effect on the tensile strength of the TE leg, but reduces its shear strength.**

For the second question, the cracks will not occur in the TiTe_2 layer?

Fig. R5 demonstrates the fracture morphology after the shear test, from which it can be seen that part of the Ti particles remains on the surface of the BiTe, and part of the BiTe is also attached to the surface of the Ti layer. Therefore, it is certain that the cracks passed through the reaction layer, but where the cracks originated? Since the TiTe_2 layer is only a few tens of nanometers thick, whether the cracks originated from TiTe_2 or BiTe, or even from the Ti layer, and at their interface, we have no way to prove it for the time being.

Fig. R5. Fracture morphology of Ti/BiSbTe joint. (a) BiSbTe side, and (b) Ti layer side.

Supplementary note: It is well known that brittle ceramics or intermetallics can even exhibit considerable ductility if the grain size is reduced to a few nanometers, which originates from the diffusional flow of atoms along the intercrystalline interfaces. (Nature, 330 (1987) 556; Scr. Metal. Et. Mater, 25 (1991) 811). Therefore, even though the interfacial reactant is brittle, it has the potential to exhibit good ductility and ensure the stabilization of the interfacial structure as long as it is thin enough and the grains are small enough. According to the TEM result, the thickness of TiTe_2 is less than 30

nm and it is composed of many finer grains. Therefore, we speculate that the crack is unlikely to originate within TiTe_2 , but rather at the interface between the two phases with high residual stress.

3. Does the loose Ti layer possess enough strength? How to ensure the leg does not break inside the Ti layer during the fabrication and service?

Response: Thanks for your comment. As shown in Fig. S14 and Table S2, the shear, compressive and tensile strength are 11.2 MPa, 138 MPa, 127.3 MPa respectively for 673 Ti. Compared to bismuth telluride, 673 Ti has only slightly lower shear strength, and the tensile strength is one order of magnitude higher than that of bismuth telluride, so the strength of Ti is sufficient for processing. If the Ti layer is damaged during the processing, there is a high probability that this external force will also cause the fracture of BiTe itself. In fact, the Ti layer, the BiTe materials, and their interfaces never cracked during the cutting (Fig. R6) and welding processes. Moreover, the module test result showed that there was no noticeable deterioration of output properties and internal resistance after 30 thermal shocks over a total service time of 360 hours at T_h of 523 K, demonstrating that the structure of module is stable and Ti layer is strong enough.

Fig. R6 The optical pictures of TE legs after cutting.

4. The authors claimed that the particle sliding occurred in the Ti layer within the temperature range from RT to 573 K, but the sliding seems imperceptible in the SEM images (Fig.4).

Response: Thanks for your comment. Due to file format conversion of the manuscript processing system, the image resolution has decreased. In fact, we can clearly see the obvious displacement between particles in the high-resolution Fig. 4 i1-k2.

Fig. 4 i1-k2. SEM images of the Ti layer at **i1-k1.** (RT) and **i2-k2.** (573 K) respectively for comparison.

Based on the formula of CTE, $\alpha = (\Delta L/L_0)/\Delta T$, where ΔL represents the change in length, L_0 represents the original length, ΔT represents the change of temperature. According to the measured CTE, when the Ti layer is heated from RT to 573 K, it can only expand by $\sim 1.8 \times 10^{-3}$ of its original length. Taking the area shown in Fig. 4 as an example, **its lateral dimension is $\sim 153 \mu\text{m}$, and the total expansion within this field of view should be $\sim 0.28 \mu\text{m}$.** However, as shown in Fig. j2, **a particle with a contour size of $\sim 20 \mu\text{m}$ has been shift by $\sim 0.32 \mu\text{m}$,** which is definitely not caused by thermal expansion. Therefore, particle sliding does indeed occur. We have labeled the sliding distances in the revised **Fig. 4.**

5. Why did not the particles continue to slid from 573 K to 623 K?

Response: Thanks for your comments. We are very sorry for not describing this in detail in the original article.

According to our analysis in the main text, particle sliding occurs during the heating process from 523 K to 573 K. However, it should be emphasized that from room temperature to 523 K, a large amount of stress has accumulated at the interface, which leads to particle sliding to release stress during the subsequent heating process. From 573 K to 623 K, although the temperature also increased by 50 K, there is no initial stress at the interface as high as before. In addition, as shown in Fig. 4b, the Young's modulus of both Ti layer and BiTe continues to decrease with increasing temperature and elastic deformation becomes easier, especially for the Ti layer. Therefore, the accumulated stress from 573 K to 623 K may be no longer sufficient to drive new sliding. We have added this discussion into the revised manuscript (lines 290-294).

6. Does the particle sliding occur in other metal layers? such as Zr or Nb, which possess

lower Young's modulus than Ti. Why did not the authors choose the two metals as the barrier layer?

Response: We gratefully appreciate your comment. We have previously explored the possibility of Nb and Zr as a barrier layer. The Ti, Nb and Zr are powders with an average particle size of 45 μm purchased from Alfa Aesar, a well-known raw material company. However, the melting point of Nb (2469 $^{\circ}\text{C}$) and Zr (1852 $^{\circ}\text{C}$) is so high that their powders cannot be compacted at 673 K. As shown in Fig. R7, after sanding using a sandpaper of 320 mesh, the surface of the Ti layer exhibits a metallic luster, while the Nb layer and Zr layer are clearly not dense enough. Furthermore, a knife can easily leave scratch marks with powder debris on their surfaces. In other words, the sintered Nb (Zr) layer possesses extremely poor mechanical properties and therefore cannot be used as barrier layers. Therefore, we did not further characterize their microstructure.

Fig. R7. The optical pictures of (a) Ti, (b) Nb, and (d) Zr surfaces after sanding using sandpaper of 320 mesh. The optical pictures of (c) Nb and (e) Zr surfaces after scratching with a knife.

7. In the manuscript, the measured output performance is inferior to the calculated values because the smaller actual temperature difference. Maybe the authors can increase the hot side temperature and measure the output performance under the same temperature difference with the calculation.

Response: Thanks for your careful reading and kind advice.

During the measurement, flexible graphite is used as the contact material between the module and the heater, while the hot-side temperature T_h is collected from the heater, which is therefore slightly higher than the actual temperature of the hot end of the module due to the thermal resistance caused by flexible graphite and multiple interfaces.

We have also attempted to change the testing position of the thermocouple by directly contacting the module with a sheet-type thermocouple and changing the contact material to silicone grease or liquid InGa alloy, ultimately increasing the tested efficiency to $\sim 6.35\%$, as shown in Fig. R8. However, this method cannot be used for

long-term testing because silicone grease will dry out over a long period of time at high temperatures, leading to a decrease in thermal conductivity, while InGa alloy will slowly invade the thermocouple, resulting in inaccurate temperature measurement. Since this work mainly focuses on the stability of modules, we did not adopt these above methods.

Fig. R8. The output voltage (a), output power (b), heat flux (c), and conversion efficiency (d) at different hot side temperatures.

As your advice, increasing the hot-side temperature and measure the output performance under the same temperature difference with the calculation, we have also seriously considered the feasibility of this approach. Theoretically, based on the relationship between Seebeck coefficient and temperature, we can deduce the actual hot-side temperature through the measured open circuit voltage, and then increase the heater temperature to bring the hot side to the target temperature. However, as mentioned in the manuscript, the gap between the calculated values and the experimental values is not only caused by the lack of temperature difference, but also by the inaccuracy of the heat flow test due to radiation leakage and other factors, which cannot be avoided at present. Therefore, it is unknown whether this method can be widely accepted by the thermoelectric community, and so we have not adopted it either.

8. There is a textual error in the Fig. S7. The (b) and (d) should be the Ti/BiTeSe instead of Ti/BiSbTe.

Response: Thanks for your careful reading. We have corrected this mistake in the revised supporting information.

In summary, thanks again for your suggestions, which have significantly improved the quality of this manuscript and we have also benefited from the discussion with you.

Reviewer #3 (Remarks to the Author):

This work aims to adopt low Yong's modulus value materials as barrier layer materials as they will result in large elastic deformation under low stress and thus relieving the interfacial stress and maintaining structural stability, even if its CTE does not match that of the TE materials. However, the interfacial stress that leads to the material fracture or the device failure is three dimensional which is more complicated than the one-dimensional Yong's modulus value in this work although it is helpful to choose barrier materials. There's still some questions need to be solved before the manuscript can be accepted.

1. For commercial Bi₂Te₃ devices modules, Ni/Ni-based materials are applied as barrier materials, however, the Yong's modulus value of Ni is ~206 GPa which is much larger than the Ti used in this work but close CTE to BiTe. Besides, it is the brittle Ni-Te compound that leads to the interface resistivity and CTE mismatch and thus final crack and failure occur above 473 K. So it is the react compound that determines the stability of the interface rather than the Yong's modulus values of the barrier materials.

Response: Thanks very much for your constructive comments! We strongly agree with you that the presence of interfacial reactants removes the direct contact between the barrier layer and the TE material, and therefore the discussion of interfacial properties needs to focus on the relationship between the reaction layer and the TE material, which we do in the interfacial resistivity section, i.e., we analyze the work function and the interfacial charge transport for TiTe₂ vs. BiTe, not Ti vs. BiTe. However, when it comes to interface stress, the situation is different, at least for this work.

For Ni/BiTe joint, the failure of the interface is indeed caused by the brittleness of the Ni-Te compound, but this is due to the increasing thickness of the reaction layer above 473 K (DOI: 10.1016/j.jallcom.2019.152731; DOI: 10.1016/j.jallcom.2016.06.207). This situation also occurs at the interface of TiAl/SKD (DOI: 10.15541/jim20170517), in which the thickness of the brittle compound CoAl continues to increase as the aging time prolongs at 848 K, causing the interface cracking. In the above two examples, the CTE of barrier layer itself is close to that of TE material, while the newly generated reactant breaks this equilibrium, and the increasing thickness of the reactant layer allows the gradual accumulation of interfacial stress, and finally cracking occurs either at the brittle material itself or at the interface. As demonstrated in reports, if the aging temperature is low (423 K for Ni/BiTe and 773 K for TiAl/SKD), the thickness of the reactive layer no longer increases rapidly and the joints will show good stability (DOI:10.1007/s11664-017-5906-x; DOI: 10.15541/jim20140378).

In the present work, unlike the above two examples, the thickness of the reaction layer TiTe₂ did not increase with time at the target temperature (250°C), and the tested bonding strength and contact resistivity are also almost constant, showing extremely high thermal stability.

Above we have discussed the stability of acquired interfaces in service, but the premise is that we have obtained an interface with low residual stress (referring to the stress that

is not sufficient to cause interface cracking). Next, we analyze the generation and release of interfacial stresses.

To simplify the problem, we only consider in-plane expansion at the interface. First, we analyze the interface between the reaction layer and the TE material. The source of interfacial stress is the inconsistency of contraction of the materials on both sides during the cooling process, and the magnitude of the force acting on the materials on both sides of the interface is independent of the material thickness theoretically. For the reaction layer, the thinner the thickness, the greater the force per unit area, the easier it is to produce deformation, if the reaction layer is very thick, the force per unit area is small, deformation becomes very difficult. Therefore, if the reaction layer is too thick to be easily deformed, it will limit the natural contraction or expansion of bismuth telluride with temperature, resulting in its fracture, since BiTe is a brittle material with poor mechanical properties.

According to the TEM result, the thickness of the reaction layer at the Ti/BiTe joint is less than 30 nm and it is composed of many finer grains. It is well known that brittle ceramics or intermetallics can even exhibit considerable ductility if the grain size is reduced to a few nanometers, which originates from the diffusional flow of atoms along the intercrystalline interfaces. (DOI: doi.org/10.1038/330556a0; DOI: 10.1016/0956-716X(91)90230-X). Therefore, even though the interfacial reactant is brittle, it has the potential to exhibit good ductility and ensure the stabilization of the interfacial structure as long as it is thin enough and the grains are small enough. This also explains why Ni/BiTe and TiAl/SKD joints maintain high stability without thickening the reaction layer. (DOI:10.1007/s11664-017-5906-x; DOI: 10.15541/jim20140378).

In fact, the situation in this article is more complex. The reaction layer is not only constrained by TE materials, but also constrained by Ti, whose CTE is different from BiTe. Since $TiTe_2$ is metallurgically bonded to both Ti and BiTe at the interface, the interfacial stresses will inevitably be transferred to them. Compared with 673Ti/BiTe joint, 1173Ti/BiTe joint undergoes cracking within BiTe (Fig. S9), and the only variable is the density of Ti in these two cases. Therefore, we believe that 673Ti bears a significant portion of the stress relief workload.

In summary, the low Young's modulus of the loose structured Ti layer and particle sliding consume a large portion of the strain confinement during the cooling process after sintering, ensuring the structural integrity of the joint. Under the action of alternating temperature not higher than 250 °C during service, the difference in thermal expansion between the two sides of the interface is small, and the interfacial stress is not enough to cause particle sliding and BiTe fracture, which are then confined in the interface. In addition, since the thickness of $TiTe_2$ no longer increase, no new stresses are introduced, and ultimately good interfacial stability is maintained.

2. Please clearly explain how the particle sliding decrease the interfacial stress in the manuscript as Fig. 1a has not shown the mechanical process.

Response: Thanks for your comment. Fig. 1a is just a schematic to summarize the

mechanism of interfacial stress release caused by loose Ti layer. We discussed the mechanism of stress release by particle sliding in page 9 (lines 280-312). In fact, we do not have direct evidence of stress release from particle sliding, and FEM is also unable to simulate particle sliding due to the inability to accurately describe the bonding state between particles. However, we can speculate from the simulation and experimental results that particle sliding does play a role in stress release.

First, from the FEM results, although the interfacial stress decreases dramatically after replacing 1173Ti with high Young's modulus by 673Ti with low Young's modulus, it is still much higher than the tensile strength of bismuth telluride, which indicates that there must be some other mechanism in place. Second, according to the in-situ SEM results, significant sliding occurred between Ti particles close to the interface, while no sliding occurred away from the interface. This is because the Ti layer far from the interface is little affected by the interfacial stress and is basically in the state of free expansion/contraction with temperature. The Ti layer close to the interface, on the other hand, is constrained by the interfacial stresses when it expands or contracts with temperature, thus generating sliding between the particles to coordinate the deformation and consume the stresses at the same time. Moreover, we discussed the irreversibility of this sliding in the article, showing that this is not an elastic deformation but a pseudo-plastic one. Further, we measured the relative displacement between the particles. According to the measured CTE, when the Ti layer is heated from RT to 573 K, it can only expand by $\sim 1.8 \times 10^{-3}$ of its original length. Taking the area shown in Fig. 4 as an example, its lateral dimension is $\sim 153 \mu\text{m}$, and the total expansion within this field of view should be $\sim 0.28 \mu\text{m}$. However, as shown in **Fig. j2**, a particle with a contour size of $\sim 20 \mu\text{m}$ has been shifted by $\sim 0.32 \mu\text{m}$, which is definitely not caused by thermal expansion. This result illustrates that the displacement is not caused by elastic deformation but by the sliding between the particles. We have labeled the sliding distances in the revised **Fig. 4**.

Fig. 4 i1-k2. SEM images of the Ti layer at **i1-k1.** (RT) and **i2-k2.** (573 K) respectively for comparison.

3. Fig. S2 displayed the contact resistivity of as-sintered Ti/BiSbTe and Ti/BiTeSe joints and also the shear strength and tensile strength of the two joints at room temperature. How about the result for Ni/BiSbTe(Se)? Or is there any reported data for compare?

Response: Thanks for your comment. It is indeed necessary to comprehensively compare the bonding characteristics of Ti and Ni with bismuth telluride. Liu et al. have studied the tensile strength of sintered Ni/BiTe joint (DOI: doi.org/10.1039/C3TA13456C). The tensile strength of Ni/p-type Bi_{0.4}Sb_{1.6}Te₃ is ~30 MPa and that of Ni/n-type Bi₂Te_{2.7}Se_{0.3} is ~16 MPa. To our knowledge, there is no relevant research on the shear strength of sintered Ni/BiTe joint. Therefore, we tested the shear strength of the joints sintered in this work. The results have been added into the revised supporting information. And the relevant description has also been added in lines 228-230.

“However, we also have to admit that due to the stronger reaction between Ni and BiTe, the shear and tensile strength of the as sintered Ni/BiTe joints are also higher than those between Ti and BiTe, as shown in **Table S1**.”

Table S1. The tensile strength and shear strength of Ni/BiTe joints at room temperature.

	Tensile strength	Shear strength
Ni/p-type BiTe	~30 MPa	~14 MPa
Ni/n-type BiTe	~16 MPa	~16 MPa

4. The authors claimed that it is the low Yong’s modulus value that leads to large elastic deformation under low stress, but in the discussion section few evidence is provided, please supply more proofs at this point.

Response: Thanks for your comment. Considering a stress-strain curve, in the elastic regime, a linear relation between stress and strain occurs. Such elastic behavior is described by Hooke’s law given by $\sigma = E\varepsilon$, where the proportionality constant E is called the modulus of elasticity or Young’s modulus. After transform the formula in the form of $\varepsilon = \sigma/E$, we can see that materials with low Young's modulus can achieve larger strains under the same stress conditions relative to materials with high Young's modulus. For the barrier layer/TE material interface, if the barrier layer has a lower Young's modulus, then the small interfacial stress has the potential to deform the barrier during cooling, thus reducing the constraint on the thermal expansion/contraction of the TE material, i.e., relieving the interfacial stress. This is the theoretical basis for our belief that low Young's modulus materials have potential to be used as barrier materials.

We said " Due to their large elastic deformation under low stress, barrier layer materials with low Young's modulus values can partly compensate for the difference in expansion or shrinkage caused by CTE mismatch, thus relieving the interfacial stress and maintaining structural stability" in the introduction part. This is just a simple scientific explanation we give based on the definition of Young's modulus. Achieving large deformation at low stress is a concrete expression of the concept of low Young's

modulus.

In the discussion section, we have presented the experimental values of the Young's modulus for 1173Ti and 673Ti in Fig. 4b, the latter is indeed lower, and it is clear from the FEM analysis that the low Young's modulus does reduce the interfacial stresses, as shown in Figs. 4c-4f. What we are concerned about here is not how much absolute deformation 673Ti undergoes due to its low Young's modulus, but rather that, compared to 1173Ti, a larger deformation (**the measured low Young's modulus itself is a criterion for large deformation under low stress**) under the same stress will alleviate the constraints during the cooling process of the materials on both sides of the interface.

In fact, the original free contraction of the material on both sides of the interface during the cooling process is limited. Not to mention that the actual deformation of the material at the joint is difficult to measure, and even if it is measured, due to the existence of the pseudo-plastic deformation mechanism of particle sliding in the 673Ti layer, it is also very difficult to determine how much of the deformation is caused by the interfacial stresses, and how much of the deformation is caused by the contraction of itself with temperature.

In summary, thanks again for your suggestions, which have significantly improved the quality of this manuscript. We have benefited from the discussion with you, and our understanding of interfacial stresses has been deepened, while we have learned a lot of issues that need to be further explored, which will be the motivation for us to conduct more in-depth research in the future.

REVIEWER COMMENTS

Reviewer #1 (Remarks to the Author):

As authors have incorporated all the comments and suggestions raised by the reviewers, this manuscript can be accepted for publication in the present form.

Reviewer #2 (Remarks to the Author):

The authors have addressed all the questions that I raised before, and I think the revised manuscript can be accepted.

Reviewer #3 (Remarks to the Author):

Thanks for the detailed response of the authors. Based on the above response and revised manuscript, the doubt of the conclusion "Low Young's Modulus and Particle Sliding alleviates interfacial stress" lacks convincing direct evidence, so considering the high requirement and large readers of Nature Communication, this manuscript may be more suitable for other journals.

(1) For the first question from Reviewer#2: "After long-time aging, why did not the Ti layer and BiTe material continue to react and the TiTe₂ layer become thicker?"

The author explains the temperature dependence of react layer thickness, while the reason why TiTe₂ layer is only nanometers, is it caused by the high reaction energy or other reasons that prevent the further reaction? This needs more explanation because it influences the stress of the interfaces.

(2) For the fourth question from Reviewer#2: "The authors claimed that the particle sliding occurred in the Ti layer within the temperature range from RT to 573 K, but the sliding seems imperceptible in the SEM images (Fig.4)."

In the response, it is hard to tell the sliding of Ti layer basing on the SEM images, and the calculated ~0.28 μm and ~0.32 μm is too close to tell the exact reason.

Response

Dear Reviewer #3,

Our point-by-point responses (in blue text) to your assessment and comments, including changes where needed (marked in blue text in the revised manuscript), are summarized below.

Overall assessment: Thanks for the detailed response of the authors. Based on the above response and revised manuscript, the doubt of the conclusion “Low Young’s Modulus and Particle Sliding alleviates interfacial stress” lacks convincing direct evidence, so considering the high requirement and large readers of Nature Communication, this manuscript may be more suitable for other journals.

Response: With existing technological means, the process of interfacial stress release cannot be directly observed when the joint is cooled from the initial state after sintering at 673 K (the almost stress-free state) to room temperature. In this article, we have provided a wealth of available direct or indirect evidence on the mechanisms of low Young's modulus and particle sliding alleviating interfacial stress.

Evidence 1:

Under the exactly same sintering process, the interface between 1173Ti and BiTe joint cracked (Fig. S11), while the interface between 673Ti and BiTe joint did not crack, which proves that the stress at the interface of the latter is less than that of the former, **which is the most direct evidence**. As to how the stress is released, it cannot be directly observed, and we can only analyze it with the help of other means. The analysis shows that the CTEs of 1173Ti and 673Ti are almost the same, and the only difference is that the Young's modulus of the latter is obviously smaller and it has a porous structure. Therefore, we next analyze the role of these two factors on stress release.

Evidence 2:

Based on the above analysis, we performed FEM analysis based on the measured mechanical properties of 1173Ti, 673Ti, and BiTe materials. The results showed that the reduction of Young's modulus can effectively relieve the interfacial stress in the joints. **After replacing 1173Ti with 673Ti, the stress sustained by BiSbTe and BiTeSe decreases significantly from 289.5 MPa and 251.4 MPa to 189 MPa and 159.4 MPa, respectively.**

It should be re-emphasized that a low Young's modulus means that the barrier material

has a high elastic deformation ability under low stresses, thus reducing the physical constraints on the TE joint, allowing the TE material to contract/expand more comfortably with temperature, thus reducing the interfacial stresses it subjected. In the field of welding and joining, the use of materials with a low Young's modulus as a stress-buffer layer is a well-known basic theory.

Evidence 3:

From the FEM results, the interface stress is still large after the joint is cooled from the initial state after sintering at 673 K (the almost stress-free state) to room temperature, so there may be another stress relief mechanism. As we said in the manuscript, **it is not feasible to directly observe the evolution of the structure near the interface from the almost stress-free state after sintering at 673 K to RT through experiments.** Therefore, the microstructural evolution of the loosely structured Ti with increasing temperature was investigated. During the heating process, **we directly observed particle sliding** and analyzed it in depth, demonstrating that the sliding is not an elastic behavior with temperature, but a pseudo-plastic deformation behavior driven by external forces. Therefore, we believe that particle sliding can effectively alleviate interfacial stress.

In summary, we have made efforts to use numerical simulation combined with in-situ experiment to demonstrate the role of low Young's modulus and particle sliding in alleviating interfacial stress.

According to our knowledge, the elaboration or discovery of any theory or mechanism in science does not always have direct evidence. As a simple example, for crystal structure, we can speculate the crystal structure by XRD or TEM based on Bragg diffraction. For TEM, what we see through the high-resolution image is only an atomic-like image, and not the direct atomic arrangement, but this does not prevent us from identifying the order of the atoms, which is also accepted by the entire scientific community, i.e., not all conclusions are supported by **direct** evidence, subject to the development of the current testing technology.

Supplementary note: The core of this work is the development of a new diffusion barrier layer that increases the stable service temperature of bismuth telluride-based modules to 523 K, a qualitative leap compared to the long-standing limit of less than 473 K since this material was developed in the 1950s, which is a major breakthrough in the field of bismuth telluride-based devices and will dramatically accelerate its application in the power generation.

Comment 1: For the first question from Reviewer#2: “After long-time aging, why did

not the Ti layer and BiTe material continue to react and the TiTe₂ layer become thicker?” The author explains the temperature dependence of react layer thickness, while the reason why TiTe₂ layer is only nanometers, is it caused by the high reaction energy or other reasons that prevent the further reaction? This needs more explanation because it influences the stress of the interfaces.

Response: It is well known that if a reaction is to be forced to take place, enough energy must first be supplied to make the free energy of the system after the reaction lower than that before the reaction.

Regarding your question “is it caused by the high reaction energy or other reasons that prevent the further reaction?”.

In this work, after sintering at 673 K for 5 min, a TiTe₂ layer with tens of nanometers thick is formed at the interface between Ti and BiTe. This indicates that at a sintering temperature of 673 K, the externally supplied thermal energy is sufficient for the reaction to take place. As to why the thickness of the reaction is only a few tens of nanometers, this is because our sintering time was only 5 min (see Experimental section for details), and the reaction process was terminated when the heating was stopped.

Since the sintering temperature of 673 K is sufficient to induce the reaction, extending the sintering time will continue to increase the thickness of the reaction layer (Fig. S8 b and d), but this will lead to an increase in contact resistivity of the n-type joint (Fig. S8 a and c), just like the case of sintering at 723 K. Besides, extending the sintering time will cause the volatilization of BiTe constituent elements, leading to a decrease in thermoelectric performance (Fig. S7). Furthermore, as we have analyzed in the last revision, the increase in the thickness of the reaction layer may lead to a mismatch in the thermal expansion of the interface, threatening the stability of the interfacial structure. In summary, extending the sintering time can lead to a decrease in TE performance and an increase in interfacial resistance, which is not a good choice. We have added this result to Fig. S7 and Fig. S8, with corresponding explanations in the main text (Lines 200-209) as follows.

“From the above, it can be seen that increasing the sintering temperature can increase the thickness of the reaction layer, which may improve the interfacial bonding strength. In addition, since the sintering temperature of 673 K is already capable of inducing interfacial reaction to take place, prolonging the sintering time may also increase the thickness of TiTe₂ layer. Therefore, we attempted to sinter the joints at 723 K for 5 min and at 673 K for 30 min respectively, but the results showed a decrease in TE performance and a significant increase in contact resistivity of the n-type joint although

the thickness of TiTe_2 layer indeed increase (**Fig. S7, Fig. S8**), which was not worth the loss. Therefore, we retained the original sintering process of 673 K and 5 min.”

An increase in the thickness of the reaction layer will lead to an increase in contact resistance and a decrease in device efficiency, which is fatal. Since the interfacial structure of the joints is stable under the existing process conditions, the bond strength is still acceptable, and the TE performance of the material are optimal, we therefore not pay too much attention to the role of different sintering processes.

Fig. S7. Fig. R2. (a1) - (d1) The thermoelectric properties of $\text{Bi}_{0.399}\text{Sb}_{1.596}\text{Pb}_{0.005}\text{Te}_3$ sintered at 623 K, 673 K, 723 K for 5 min, and 673 K for 30 min. (a2) - (d2) The thermoelectric properties of extruded n-type BiTe re-sintered at 673 K, 723 K for 5 min and 673 K for 30 min.

Fig. S8. Contact resistivity curves (a) (c) and BSE images (b) (d) of Ti/BiSbTe and Ti/BiTeSe joints sintered at 673 K for 30 min. Contact resistivity curves (e) (f) and BSE images (g) (h) of Ti/BiSbTe and Ti/BiTeSe joints sintered at 723 K for 5 min.

Comment 2: For the fourth question from Reviewer#2: “The authors claimed that the particle sliding occurred in the Ti layer within the temperature range from RT to 573 K, but the sliding seems imperceptible in the SEM images (Fig.4).”

In the response, it is hard to tell the sliding of Ti layer basing on the SEM images, and the calculated $\sim 0.28 \mu\text{m}$ and $\sim 0.32 \mu\text{m}$ is too close to tell the exact reason.

Response: The response to Reviewer #2 was “According to the measured CTE, when the Ti layer is heated from RT to 573 K, it can only expand by $\sim 1.8 \times 10^{-3}$ of its original length. Taking the area shown in Fig. 4 as an example, its lateral dimension is $\sim 153 \mu\text{m}$, and the total expansion within this field of view should be $\sim 0.28 \mu\text{m}$. However, as shown in Fig. j2, a particle with a contour size of $\sim 20 \mu\text{m}$ has been shift by $\sim 0.32 \mu\text{m}$, which is definitely not caused by thermal expansion.”

Here, what we need to distinguish is not the difference between $\sim 0.28 \mu\text{m}$ and $\sim 0.32 \mu\text{m}$. The value of $\sim 0.28 \mu\text{m}$ refers to the amount of expansion in an area with a width of $153 \mu\text{m}$. Correspondingly, a region with a width of $\sim 20 \mu\text{m}$ should have an expansion of **$\sim 0.037 \mu\text{m}$, which is an order of magnitude lower than the particle sliding distance of $\sim 0.32 \mu\text{m}$.** Therefore, we said that “a particle with a contour size of $\sim 20 \mu\text{m}$ has been shift by $\sim 0.32 \mu\text{m}$, which is definitely not caused by thermal expansion.”

REVIEWERS' COMMENTS

Reviewer #3 (Remarks to the Author):

Thanks the authors for their reponse. The concerns have been addressed in the revisions.

November 17, 2023

Dear Editor and Reviewer #3,

Thank you very much for your recognition of our work. We have changed the format and expression of the manuscript and supporting files according to the Editor's request.